

# The vapor pressure over nano-crystalline ice

Mario Nachbar[1,2], Denis Duft[2] and Thomas Leisner[1,2]

[1]Institute of Environmental Physics, University of Heidelberg, Im Neuenheimer Feld 229, 69120 Heidelberg, Germany

[2]Institute of Meteorology and Climate Research, Karlsruhe Institute of Technology – KIT, Hermann-von-Helmholtz-Platz 1, 76344 Eggenstein-Leopoldshafen, Germany

*Correspondence to*: Mario Nachbar (Mario.nachbar@kit.edu)

**Abstract.** Crystallization of amorphous solid water (ASW) is known to form nano-crystalline ice. The influence of the
nanoscale crystallite size on physical properties like the vapor pressure is relevant for processes where crystallization of amorphous ices occurs e.g. in interstellar ices or cold ice cloud formation in planetary atmospheres, but up to now not well understood. Here, we present laboratory measurements on the saturation vapor pressure over nano-crystalline ice between 135 K and 190 K. Below 160 K, where nano-crystalline ice is known to be metastable for extended periods, we obtain a saturation vapor pressure that is 100 % to 200 % higher compared to stable hexagonal ice. This elevated vapor pressure is in
striking contrast to the vapor pressure of stacking disordered ice which is expected to be the prevailing ice polymorph at these temperatures with a vapor pressure at most 18 % higher than that of hexagonal ice. This apparent discrepancy can be reconciled by assuming that nanoscale crystallites with mean diameter between 7 nm and 19 nm form in the crystallization process of ASW. The high curvature of these nano-crystallites results in a vapor pressure increase which can be described by the Kelvin equation. Our measurements show, that at temperatures up to 160 K, ASW is the first solid form of ice deposited
from the vapor phase and that nano-crystalline ice forms thereafter by crystallization within the ASW matrix. The size of the nano-crystallites remains stable for hours below 160 K and thus nano-crystalline ice may be regarded as an independent phase for many atmospheric processes below 160 K. We parameterize the vapor pressure of nano-crystalline ice using a constant Gibbs free energy difference of $(982 \pm 182)$ Jmol$^{-1}$ relative to hexagonal ice.

## 1 Introduction

It is well-known that the crystallization process of amorphous solid water (ASW) below about 160 K forms nano-crystalline ice with crystallite diameters between 5 to 40 nm: Using electron diffraction, Jenniskens and Blake (1996) observed crystal diameters of 10 nm to 15 nm between 150 K and 160 K and Kumai (1968) reported diameters of 5 nm to 30 nm at 113 K to 143 K. Dowell and Rinfret (1960) used X-ray diffraction and observed grain sizes of about 40 nm. Crystallization of the high pressure ices II, IV, V and IX has been shown to produce nano-crystalline ice as well (Arnold et al., 1968; Kuhs et al., 1987).
This nano-granular structure may have significant effects on the properties of the ice polymorph. For example, Johari and Andersson (2015) attributed a reduction in the measured thermal conductivity of ice crystallized from ASW to enhanced phonon scattering at stacking faults and grain boundaries of the crystallites. Furthermore, the nano-crystallites might impact the vapor pressure over the ice phase, but to the best of our knowledge, this effect has not been quantified yet.

Below 160 K, nano-crystalline ice is stable for several hours (Hansen et al., 2008) and thus its vapor pressure is of relevance
for atmospheric processes occurring at these conditions, e.g. cloud formation processes in the terrestrial mesosphere or on other planets like Mars. At temperatures below 160 K, however, only a limited number of desorption rate measurements of ice crystallized from ASW using quadrupole mass spectrometers and quartz crystal microbalances is available, which may be used to calculate the saturation vapor pressure over the ice phase (Brown et al., 1996; Bryson et al., 1974; Fraser et al., 2001; La Spisa et al., 2001; Sack and Baragiola, 1993; Smith et al., 2011; Speedy et al., 1996). Measuring water vapor



desorption rates at such low temperatures is a challenging task and these measurements reveal large discrepancies among each other. This situation points to the need for high quality saturation vapor pressure measurements of nano-crystalline ice crystallized from ASW.

In this work, we report the vapor pressure of nano-crystalline ice samples deposited from the gas phase below 160 K in a temperature range between 135 K and 190 K using two independent and complementary experimental setups. One setup is based on a technique for measuring absolute saturation vapor pressures using the growth of trapped nanoparticles at isothermal conditions as a sensitive probe at temperatures between 135 K and 160 K. This setup is briefly described in Sect. 2.1. In order to extend the range to temperatures around 190 K, for which the vapor pressure of crystalline ice is established

within a few percent, we also report results from an independent more conventional setup. It allows to measure the relative vapor pressure of water ice samples with respect to hexagonal ice $I_h$ using temperature ramping in the range between 166 K and 190 K and is detailed in Sect. 2.2. In Sect. 3, we present our results from both setups. In Sect. 4, we discuss our results and compare them with the literature.

## 2 Experimental

### 2.1 Isothermal vapor pressure measurements using MICE-TRAPS (T=135 K - 160 K)

The molecular flow ice cell within the trapped reactive atmospheric particle spectrometer (MICE-TRAPS) (Duft et al., 2015; Meinen et al., 2010) has been used previously to investigate adsorption and nucleation of $CO_2$ on levitated small metal-oxide nanoparticles at low temperatures (Nachbar et al., 2016). In this work, we expose the nanoparticles to a flow of water molecules originated from temperature controlled ice covered sample surfaces. We utilize the growth of the nanoparticles as

a sensitive probe for the sublimation rate of water molecules and thus the saturation vapor pressure over the ice covered sample surfaces. In the following, we briefly recall the experimental setup and introduce the experimental procedure applied in this study.

We generate single charge sub-4 nm radius Silica ($SiO_2$, $\rho = 2.3 \, \mathrm{kg \, m^{-3}}$) and iron oxide ($Fe_2O_3$, $\rho = 5.2 \, \mathrm{kg \, m^{-3}}$) particles in a non-thermal, low pressure microwave plasma particle source. The nanoparticles are transferred into TRAPS, a low

pressure vacuum apparatus where they are size selected and stored in MICE, which is a combination of a linear ion trap and a supersaturation cell operating in the molecular flow regime (Duft et al., 2015). An illustration of the radial cross section of MICE is shown in Fig. 1.

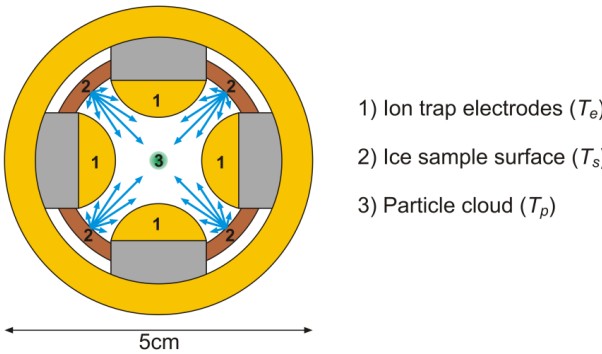

1) Ion trap electrodes ($T_e$)

2) Ice sample surface ($T_s$)

3) Particle cloud ($T_p$)

5cm

**Figure 1: Radial cross section of MICE.**

In MICE, the charged nanoparticles are trapped in the center between the four quadrupole ion trap electrodes (1), where the nanoparticles are exposed to an adjustable supersaturation of $H_2O$ molecules. This is achieved by temperature controlled sublimation from two sets of ice covered gold surfaces which are held at two distinct temperatures and which completely



surround the levitated particles. These surfaces are the ion trap electrodes held at temperature $Te$ and the ice covered sample surfaces in between the electrodes, which can be heated to an offset temperature $Ts$. The temperature of the nanoparticles equilibrates by collisions with a Helium background gas and can be calculated as described in (Duft et al., 2015). Before the start of each measurement series, a several micro meter thick layer of ice is deposited from the gas phase at variable sample

surface temperatures between 95 K and 160 K with a deposition rate between 1 nm s$^{-1}$ and 10 nm s$^{-1}$. Deposition is performed while evacuating the TRAPS chamber and water vapor is provided from a water reservoir containing Nanopure™ water. After deposition, the ice-covered electrodes and additional surfaces are set to the desired temperature such that saturation in excess of S=1000 is established. This is achieved by setting a temperature difference of 20 K or more between the sample surfaces and the cold electrodes. At such high S-values, the critical saturation for ice nucleation is exceeded by

far and the particles will nucleate ice and continue to grow. The particle growth is monitored by extracting small fractions of the trapped particle population from MICE at periodic residence times and directly measuring the particle mass using a time-of-flight mass spectrometer. The high temperature difference between the electrodes and the warm surfaces has the beneficial effect that sublimation from the cold electrodes is at least 10$^3$ times less than from the warmer sample surfaces and can be neglected. Accordingly, the sublimation rate of H$_2$O molecules from the warm sample surfaces held at $T_s$ determines

the ice particle growth rate. The ice on these surfaces constitutes the sample of interest, which for each measurement has been kept at a constant temperature between 135 K and 160 K. We use the measured particle mass growth rates to calculate the temperature dependent sublimation rate from the sample surfaces in MICE and convert them directly into a saturation vapor pressure. Example measurements, a detailed description of the nanoparticle growth rate model and the data analysis are given in Appendix A.


## 2.2 Relative vapor pressure measurements during temperature ramping with an ionization gauge (T=166 K – 190 K)

In order to extend the saturation vapor pressure measurements to temperatures above 160 K, we used an additional experimental setup and measured the relative vapor pressure difference of metastable crystalline ice and ice I$_h$ between 166 K and 190 K. The setup consists of a vacuum chamber with a $5 \cdot 10^{-9}$ mbar residual gas pressure which encloses a

temperature-controlled flat copper surface (95 cm²) onto which a roughly 15 μm thick ice sample of interest is deposited from the gas phase at a deposition rate of about 8 nm s$^{-1}$. Water vapor is provided from a water reservoir containing Nanopure™ water that has been subject to several freeze-pump-thaw cycles to remove dissolved gases from the liquid prior to deposition. In this setup, crystalline ice is produced using the same procedure as with the MICE-TRAPS setup, either via deposition of ASW at 100 K followed by crystallization during warm-up or by direct deposition at 150 K. As a reference

sample, hexagonal ice is produced by condensation of liquid water on the target at about 270 K and subsequent freezing of the liquid water at about 260 K. Following ice formation, the sample temperature is set to 150 K at which point cooling is turned off to allow for a slow warm-up (~0.5 K min$^{-1}$). The vapor pressure in the chamber is recorded as function of the sample temperature between 166 K and 190 K using a hot-cathode ionization gauge. A Quadrupole Mass Spectrometer is used to confirm that no gases other than H$_2$O bias the pressure readout. To avoid the systematic errors occurring in absolute

vapor pressure measurements with a hot cathode ionization gauge, we report only the relative vapor pressure of the low temperature deposited samples with respect to the hexagonal ice sample using otherwise identical experimental procedures. For a more detailed description of this setup the reader is referred to Appendix B.

## 3 Results

Isothermal saturation vapor pressure measurements were performed with MICE-TRAPS in the temperature range between

133 K and 160 K and non-isothermal measurements using the hot ionization gauge setup were performed with a temperature




ramp of 0.5 Kmin$^{-1}$ between 166 K and 190 K. The results are shown in Fig. 2 relative to the saturation vapor pressure of hexagonal ice $p_{sat}^h$ taken from the parameterization of Murphy and Koop (Murphy and Koop, 2005), which is expected to be accurate to within 1% at the temperatures under investigation. At the beginning of each MICE-TRAPS experiment, water ice films were deposited on the surfaces in MICE either at 95 K, 140 K or 160 K. After ice deposition was completed, the

temperature of the ice sample of interest was set to the desired temperature and isothermal measurements as described in Sect. 1.1 and Appendix A were carried out. The results are presented in Fig. 2 by the blue diamonds, green triangles and red squares. Temperature error bars are of the same size as the data points ($\Delta T = 0.2$ K - 0.4 K). The blue diamonds show the results of a series of six measurements performed using a single ASW film deposited at 95 K with the arrow indicating the chronology. The series started at 133.4 K with the freshly deposited film followed by a repeated sequence of setting the

desired sample temperature and 20 minutes of thermalization followed by the measurement of particle growth at constant temperature. For this set of measurements, we observe a sharply decreasing relative vapor pressure between 133 K and 140 K (the first 4 data points) which levels off to the saturation vapor pressure obtained for the samples deposited at 140 K and 160 K (green triangles and red squares, respectively). We interpret this behaviour as the thermally activated crystallization of ASW. From our data we estimate the crystallization constant $\tau$ to be about 25 min at 140 K, which is in agreement with

previously reported temperature dependent crystallization constants and times (Dowell and Rinfret, 1960; Sack and Baragiola, 1993; Smith et al., 1996; Smith et al., 2011).

Above 140 K, the saturation vapor pressure is found to be independent of the deposition temperature, suggesting that ice deposited between 140 K and 160 K forms the same ice polymorph as ice crystallized from ASW. Between 135 K and 160 K the vapor pressure of this ice polymorph is elevated by a factor between 2 and 3 with respect to ice hexagonal ice I$_h$.

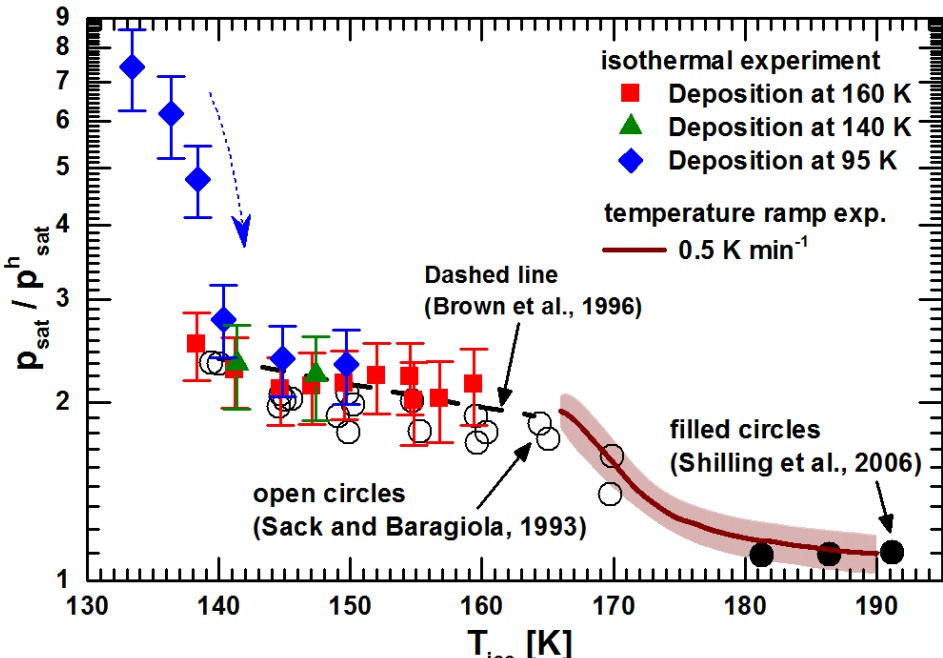


**Figure 2: Measured relative saturation vapor pressure of low temperature deposited ices with respect to ice I$_h$ as a function of temperature. Green triangles (▲) and red squares (■) denote isothermal vapor pressure measurements of ice samples deposited at 140 K and 160 K, respectively. Blue diamonds (◆) represent a series of subsequent isothermal measurements using a single ice film deposited at 95 K with the arrow indicating the chronology. The brown line is the combined experimental result for the non-**
**isothermal relative vapor pressure measurements of all ices deposited below 150 K (including a shaded interval of uncertainty). For comparison, we show data obtained for crystalline ice from the literature works (see text in figure).**



At higher temperatures, between 166 K and 190 K, the saturation vapor pressure of vapor deposited ice was measured using the hot ionization gauge setup. As reported above for the MICE experiment, we find that ice crystallized from ASW after deposition at 100 K and ice deposited at 150 K do not show any significant difference in vapor pressure in this temperature range (see Appendix B). The average of the relative vapor pressures obtained for all runs performed with the hot ionization gauge setup is shown as a brown line in Fig. 2 with the shaded area indicating one standard deviation. Above 180 K, our measurements show an only slightly elevated vapor pressure between 3 % and 30 % above that of hexagonal ice. At lower temperature, however, the measured normalized vapor pressure of the crystalline ice phase increases and connects well to the vapor pressure measured with the MICE-TRAPS setup at 160 K. It is noteworthy that for an ice layer thickness of several micrometer as was studied here we do not see any difference between substrate materials (gold in MICE-TRAPS and copper in the ionization gauge setup).

## 4 Discussion

### 4.1 Comparison to literature data

We reviewed and partially re-analyzed the limited number of available literature data on the desoprtion rate of metastable ice below about 170 K in order to compare them with our measurements. These measurements typically employed a quadrupole mass spectrometer (QMS) and/or a quartz crystal microbalance to measure desorption rates. Desorption rates can be used to infer saturation vapor pressures under the well supported assumption that the sticking coefficient for water molecules on water ice is unity at these temperatures (Batista et al., 2005; Brown et al., 1996; Gibson et al., 2011; Kong et al., 2014). Measuring water vapor desorption rates at temperatures under investigation is a challenging task and previous experiments were influenced by contamination issues, showed very large degree of scattering in the data or yielded unphysically low vapor pressures below that of ice $I_h$ (Bryson et al., 1974; Fraser et al., 2001; La Spisa et al., 2001). Sack and Baragiola (1993) carefully avoided contributions of water molecules from external sources by shielding the ice sample with cold surfaces held at 12 K and measured the desorption rate at a constant temperature. We converted their data representing ice crystallized from ASW (Fig. 2 in (Sack and Baragiola, 1993)) to vapor pressures normalized to ice $I_h$ and reproduce them as open circles in Fig. 2. Brown and co-workers measured temperature dependent desorption rates with a QMS (Brown et al., 1996). We converted their parameterized data to normalized vapor pressure values and show them as a black dashed line in Fig. 2. Both results match our measurements very well. The authors of both articles do not discuss potential causes of the measured elevated vapor pressure with respect to ice $I_h$. However, in the case of (Brown et al., 1996) a temperature error of 2 K is assumed , large enough to make their results agree with the vapor pressure of ice $I_h$ within the limits of error. Sack and Baragiola (1993) do not discuss the temperature uncertainty of their sample and the accuracy of their measurements in detail. Comparing the results of (Brown et al., 1996) and (Sack and Baragiola, 1993) to the results of the MICE-TRAPS experiments below 160 K, we assume that the reported desorption rates in both publications are rather accurate and support our measurements of an elevated vapor pressure with respect to ice $I_h$ between a factor of 2 and 3.

Hexagonal ice is the lowest energy phase of solid water below the freezing point under typical terrestrial atmospheric conditions. The overall thermodynamic model of ice $I_h$ is consistent and is supported by data obtained from a variety of different experiments (e.g. Feistel and Wagner, 2006, 2007; Murphy and Koop, 2005). Below about 200 K however, water may be encountered in the metastable cubic form ice $I_c$ (e.g. Hobbs, 1974). Recently, studies using diffraction measurements and numerical simulations showed that samples were not composed of pure cubic ice, but rather exhibited crystalline sequences of cubic ice interlaced with sequences of hexagonal ice (e.g. Hudait et al., 2016; Kuhs et al., 2012; Lupi et al., 2017; Malkin et al., 2015; Murray et al., 2015; Shallcross and Carpenter, 1957; Thürmer and Nie, 2013). This ice polymorph has been termed stacking disordered ice $I_{sd}$. It is metastable and eventually transforms to the stable ice $I_h$. Cubic ice and hexagonal ice are both based on stacked layers of water molecules in sixfold symmetry, differing only in the stacking





sequence of these layers. Hence, most physical properties of cubic and hexagonal ice are quite similar (Kuhs et al., 2012). Consequently, the vapor pressure of ice I$_{sd}$ is expected to be only slightly higher compared to ice I$_h$. In general, the higher vapor pressure $p_{sat}^m$ of such a metastable ice polymorph compared to the vapor pressure of hexagonal ice $p_{sat}^h$ is reflected by a Gibbs free energy difference $\Delta G_{m \to h}(T)$, which can be separated into an enthalpy and an entropy contribution according to:

$$\frac{p_{sat}^m}{p_{sat}^h} = exp\left(\frac{\Delta G_{m \to h}}{RT}\right) \quad with: \quad \Delta G_{m \to h} = \Delta H_{m \to h} - T\Delta S_{m \to h} \tag{1}$$

Under the assumption that the entropy difference $\Delta S_{sd \to h}$ is close to zero (e.g. (Tanaka, 1998; Tanaka and Okabe, 1996)), $\Delta G_{sd \to h}$ equals $\Delta H_{sd \to h}$. The transformation of ice I$_{sd}$ to ice I$_h$ at temperatures above 180 K has been studied extensively with differential scanning calorimetry (DSC) (e.g. Handa et al., 1986; Mayer and Hallbrucker, 1987; McMillan and Los, 1965; Sugisaki et al., 1968). These studies determined the enthalpy difference $\Delta H_{sd \to h}$ between the two ice phases to be in the range of 20 J mol$^{-1}$ to 180 J mol$^{-1}$, which according to Eq. (1) correspond to a vapor pressure difference of 1 % to 18 % between 130 K and 190 K. This is in agreement with direct vapor pressure measurements (black dots in Fig. 2) revealing a difference of about 10 % (Shilling et al., 2006) and our results between 180 K and 190 K, but is in striking contrast to our data below 170 K. In the following we will show, that the observed elevated vapor pressure below 170 K can be attributed to the formation of nano-scale grains formed upon the crystallization of ASW.

**4.1 The effect of nano-crystalline ice on the vapor pressure**

It is well-known that the crystallization process below 166 K of ASW as well as the high pressure ices II, IV, V and IX forms nano-crystalline ice (Arnold et al., 1968; Backus and Bonn, 2004; Dowell and Rinfret, 1960; Jenniskens and Blake, 1996; Kondo et al., 2007; Kuhs et al., 1987; Kumai, 1968). The formation of nano-crystallites is believed to occur by nucleation of ice embryos followed by their diffusional isotropic 3-dimensional growth within the remaining ASW matrix until all amorphous water is transformed to crystalline ice. At low temperatures, the interplay of ice nucleation and ice growth leads to nanoscale crystallites (e.g. Backus and Bonn, 2004; Kondo et al., 2007). A nano-crystallite exhibits a large surface energy to volume energy ratio resulting in an increased vapor pressure above its surface. This vapor pressure increase is described by the Kelvin equation which at the same time corresponds to the vapor pressure increase over a macroscopic surface composed of spherical nano-grains:

$$ln\left(p_{sat}^{nano}/p_{sat}^{cryst}\right) = 4 \cdot v \cdot \sigma / k \cdot T \cdot d_{grain} \tag{2}$$

Equation (2) describes the vapor pressure increase over a curved surface of spherical nano-grains with a grain diameter $d_{grain}$ consisting of crystalline ice with a bulk vapor pressure $p_{sat}^{cryst}$. Here, $k$ is the Boltzmann constant, T is the temperature, $v$ is the molecular volume, and $\sigma$ is the ice-vapor surface tension of the crystalline ice. We assume that the crystalline nano-grains are composed of ice I$_{sd}$ as supported by model studies (Lupi et al., 2017) and x-ray diffraction experiments (Morishige et al., 2009). Since the surface tension of hexagonal and cubic ice are assumed to be very similar, we used the surface tension parametrization of hexagonal ice for ice I$_{sd}$ ($\sigma_{sd} = 0.001 \cdot (141 - 0.15 \cdot T[K])$ [N m$^{-1}$] (Hale and Plummer, 1974)) and assumed an uncertainty of 10 %. We inferred $p_{sat}^{sd}$ using Eq. (1) and assumed a free energy difference of ice I$_{sd}$ to ice I$_h$ of 20 J mol$^{-1}$ to 180 J mol$^{-1}$ in order to calculate grain diameters needed to explain the observed elevated vapor pressure found in this work using Eq. (2). The results for the grain diameters calculated from the MICE-TRAPS data (black squares) and from the relative vapor pressure measurements in the temperature ramp experiment (brown line, with shaded interval of confidence) are shown in Fig. 3. Below 160 K, estimated grain size diameters are in the range between 7 nm and 19 nm. According to our measurements, the crystal size does not depend on the formation temperature below 160 K and remains constant over a typical measurement period of 10 hours.





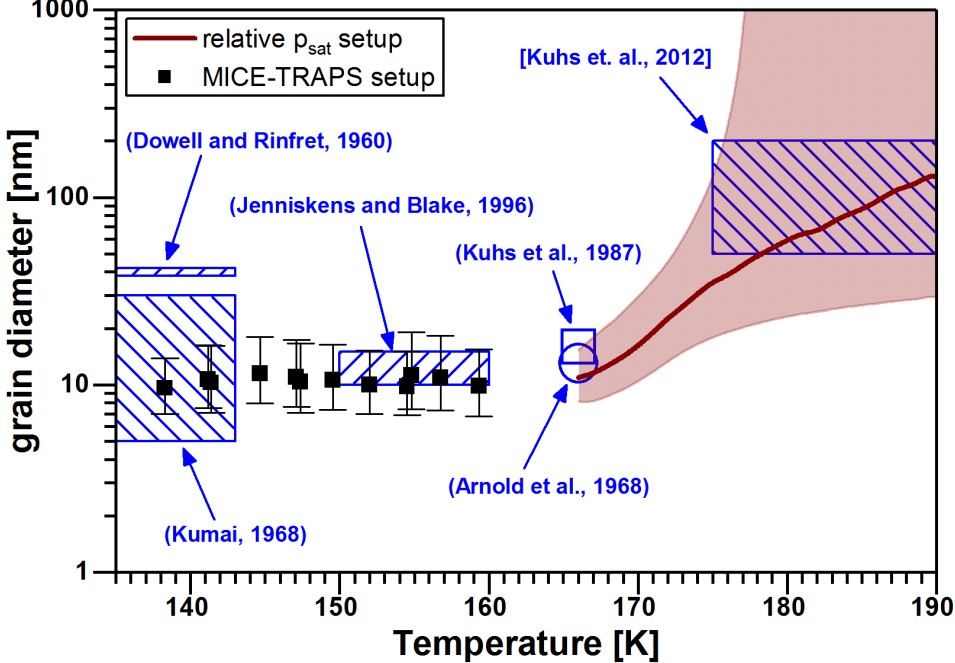

**Figure 3: Calculated nano-crystallite diameters as a function of temperature in ice crystallized from ASW. The black squares (■) represent the results of the MICE-TRAPS measurements and the brown curve with shaded confidence interval the results of the relative vapor pressure measurements. The results are compared to crystal diameters reported in the literature (blue shaded areas and dots).**

Small grain sizes like this have previously been observed after crystallization of vapor-deposited ASW: Jenniskens and Blake (1996) observed crystal diameters of 10 nm to 15 nm between 150 K and 160 K and Kumai (1968) reported diameters of 5 nm to 30 nm at 113 K to 143 K using electron diffraction. Dowell and Rinfret (1960) used X-ray diffraction and observed grain sizes of about 40 nm. The crystallization process of deuterated water from the high vapor pressure ices II, IV, V and IX has been examined in several studies with neutron powder diffraction: Kuhs et al. (1987) observed mean particle diameters of 16 nm and Arnold et al. (1968) reported mean crystal sizes of 13 nm. The reported crystal grain diameters are indicated in Fig. 3 by the blue areas and dots. These measurements (except for (Dowell and Rinfret, 1960)) agree well with our grain diameter calculations. The non-isothermal relative vapor pressure measurements suggest that above 165 K, crystal growth is effectively activated by sublimation and re-condensation at the crystal surface or by local water molecule migration across grain boundaries, which is accompanied with a decrease of the $p_{sat}/p_{sat}^h$ ratio. This conclusion is supported by a study of Hansen and co-workers who measured the grain sizes of deuterated ice with neutron powder diffraction and small angle neutron scattering as function of temperature (Hansen et al., 2008). They report mean crystal diameters between 20 nm and 25 nm after the crystallization process with crystal sizes being stable for hours up to temperatures of about 160 K followed by crystal growth at higher temperatures. At temperatures between 175 K and 190 K, Kuhs et al. (2012) observed crystal sizes by SEM imaging and neutron diffraction between 50 nm and 200 nm. The crystal sizes increased with temperature and match the diameters calculated from our measurements in this temperature range.

In order to calculate crystal diameters, we assumed that the crystallites are composed of ice $I_{sd}$ and that this ice polymorph is described by a temperature independent Gibbs free energy difference $\Delta G_{sd \to h}$ of 20 J mol$^{-1}$ to 180 J mol$^{-1}$. These values were determined at temperatures above 180 K and a different free energy difference below 160 K would lead to a change in calculated crystallite sizes. An increase of defects beyond stacking faults in the ice $I_{sd}$ polymorph with decreasing temperature might cause an increase in $\Delta G_{sd \to h}$ (Hudait et al., 2016). However, we used the vapor pressure measurements below 160 K and calculated the Gibbs free Energy difference of the nano-crystalline ice with respect to ice $I_h$ which turned





out to be a constant value of $\Delta G_{n \to h} = (982 \pm 182)$ Jmol$^{-1}$. A significant change of $\Delta G_{sd \to h}$ with decreasing temperature should be directly seen in a slope of the $\Delta G_{n \to h}$ values which is not observed. Because of that and since our calculations of crystal sizes match previously reported values, we conclude that the increased vapor pressure below about 170 K is of morphological origin and can be explained solely by the well-established formation of nanoscale grains. The grains are

stable for several hours below 160 K and thus nano-crystalline ice can be regarded as an independent phase at these temperatures, which is described by a constant Gibbs free energy difference of $\Delta G_{n \to h} = (982 \pm 182)$ Jmol$^{-1}$ in respect to hexagonal ice.

Since deposition between 140 K and 160 K as well as crystallization of ASW deposited at 95 K and 100 K lead to the same nano-crystallite sizes, it is very likely that ice deposition up to 160 K proceeds by an initial deposition of ASW followed by

rapid crystallization to nano-crystalline ice. This conclusion is supported by the work of Chonde and co-workers. They used deposition rates comparable to our work and observed non-porous ASW immediately after deposition at 140 K (Chonde et al., 2006). At temperatures above 140 K, we cannot observe the crystallization process after deposition of ASW with the MICE-TRAPS setup since the time needed to perform the first experimental run exceeds the crystallization time at these temperatures.

It is well known, that ASW might be deposited in a porous form, which depends on deposition angle, rate and temperature (Dohnalek et al., 2003; Hill et al., 2016; Kimmel et al., 2001a; Kimmel et al., 2001b; Kouchi et al., 1994; Mayer and Pletzer, 1986; Mitterdorfer et al., 2014; Raut et al., 2007; Stevenson et al., 1999). Deposition of ASW at temperatures between 90 K and 110 K revealed either small degrees of porosity (Brown et al., 1996; Chonde et al., 2006) or were non-porous (Kimmel et al., 2001b; Stevenson et al., 1999). Thus, reports of the porosity of ASW deposited at conditions comparable to our studies

are inconsistent and we cannot exclude a small degree of porosity of our ASW samples. However, due to the fact that independent of deposition temperature the same crystalline ice polymorph forms, we conclude that either all our ASW samples are non-porous or that any porosity of the ASW sample deposited at 95 K and 100 K has no influence on the ice grain sizes formed during crystallization. The latter is supported by the observation of a strong decrease of density of micro-porous ASW at annealing temperatures above 100 K with complete absence of micro-pores above temperatures of 140 K

(Hill et al., 2016; Kimmel et al., 2001b; Raut et al., 2007).

## 4 Conclusions

We present saturation vapor pressure measurements of water ices deposited from the vapor phase at temperatures below 160 K using two independent and complementary experimental approaches. One experiment is based on a novel technique using nanoparticles as sensitive probes for isothermal absolute sublimation rate measurements (135 K – 160 K), and a more

conventional setup uses a hot ionization gauge for relative vapor pressure measurements during a temperature ramp experiment (166 K – 190 K).

Our vapor pressure measurements below 160 K show a 2 to 3 times higher saturation vapor pressure compared to ice I$_h$. These results are consistent with previously reported measurements (Brown et al., 1996; Sack and Baragiola, 1993). The observed high vapor pressure is quantitatively explained with the high surface energy to volume energy ratio of nano-scale

crystallites (Kelvin effect). A transition in the vapor pressure data above 165 K is explained by the thermally activated relaxation of nano-crystalline to stacking disordered ice of larger grain size, thereby gradually reducing the Kelvin effect. Above 180 K, the measured saturation vapor pressure levels-off at values representative for ice I$_{sd}$ at these temperatures.

From the fact that the same nano-crystalline ice polymorph forms by vapor deposition below 160 K and by crystallization from ASW, we conclude that even at temperatures as high as 160 K, amorphous ice is the initial phase formed by ice

deposition from the vapor, prior to crystallization. This is important for ice cloud processes which occur below 160 K as it implies that ice nucleation rates at these temperatures are dominated by the properties of ASW rather than those of





crystalline ice. After crystallization, however, ice growth processes are described by the properties of nano-crystalline ice. The mean crystallite size of 7 nm to 19 nm in diameter determined in this work is stable for hours below 160 K. We therefore propose considering nano-crystalline ice as an independent phase in ice cloud processes below 160 K. For practical reasons, we provide a parameterization for the saturation vapor pressure over this ice polymorph and suggest it to be used in

a temperature range where the transformation time to microscopic crystal sizes is long compared to the processes involved. Below 160 K, $p_{sat}^{nano}$ may be parameterized using a constant Gibbs free energy difference of $\Delta G_{n \to h} = (982 \pm 182)$ Jmol$^{-1}$ relative to the well-established parameterization for hexagonal ice (Murphy and Koop, 2005).

Our findings are of importance for cloud processes in the middle atmospheres of planets. For instance, water ice clouds are frequently observed in the middle atmosphere of Mars (Guzewich et al., 2013; Vincendon et al., 2011) with temperatures

commonly falling below 160 K (Maltagliati et al., 2011). In the terrestrial atmosphere, Noctilucent Clouds form at the high latitude summer mesopause (Rapp and Lübken, 2004) with temperatures falling to 120 K on average (Lübken et al., 2009) with extremes down to 100 K (Rapp et al., 2002). In addition, the vapor pressure of nano-crystalline ice is important for modelling $H_2O$ adsorption and desorption processes in interstellar environments and water residence times on interstellar grains (Fraser et al., 2001).

**Data availability**

All data is available on request from the corresponding author.

**Appendix A: Nanoparticle growth model**

In general, the mass growth rate *dm/dt* of ice particles exposed to water vapor can be expressed as the difference of the water vapor deposition rate ($k_{dep}$) on the particle surface and the sublimation rate ($k_{sub}$) from the particle surface:

$$\frac{dm}{dt} = \left[ k_{dep} - k_{sub} \right] \cdot m_{H_2O} \qquad (A1)$$

with $m_{H_2O}$ being the mass of one water molecule. To avoid later complications and uncertainties due to sublimation from the particle surface, MICE was operated in the experiments presented here at conditions of very high supersaturation of S=1000 and above where $k_{dep} \gg k_{sub}$. Under these conditions, sublimation from the particle surface can be neglected. In MICE, high supersaturation is achieved by setting a temperature difference of 20 K or more between the sample surfaces and the cold

electrodes. This has the beneficial effect that sublimation from the electrodes is at least $10^3$ times less than from the sample surfaces and can be neglected, which simplifies the calculation of the deposition rate. Under the well-supported assumption of a sticking probability of unity for water molecules on water ice under the experimental conditions employed here (Batista et al., 2005; Brown et al., 1996; Gibson et al., 2011; Kong et al., 2014), Eq. (A1) can be expressed in terms of the saturation vapor pressure $p_{sat,s}$ over the ice sample surfaces, yielding for the particle mass growth rate:

$$\frac{dm}{dt} = A_c(t) \frac{F_s v_{th,s}}{4kT_s} \cdot m_{H_2O} \cdot p_{sat,s} \qquad (A2)$$

Here, $A_c = 4\pi \left( r_p + r_{H_2O} \right)^2$ is the effective particle surface area assuming spherical ice particles, $v_{th,s} = \sqrt{8kT_w / \pi m_{H_2O}}$ is the mean thermal velocity of vapor phase molecules at temperature $T_s$, and $F_s = 0.274 \pm 0.008$ is the solid angle weighting factor of the sample surfaces as seen from the particle location, which was determined by numerical calculation based on the geometry of MICE.

The upper panel of Fig. A1 shows the measured particle mass as a function of trapping time in MICE for three exemplary measurements with sample-surface temperatures of 147.4 K, 149.7 K (particle material: $Fe_2O_3$) and 154.8 K (particle material: $SiO_2$).



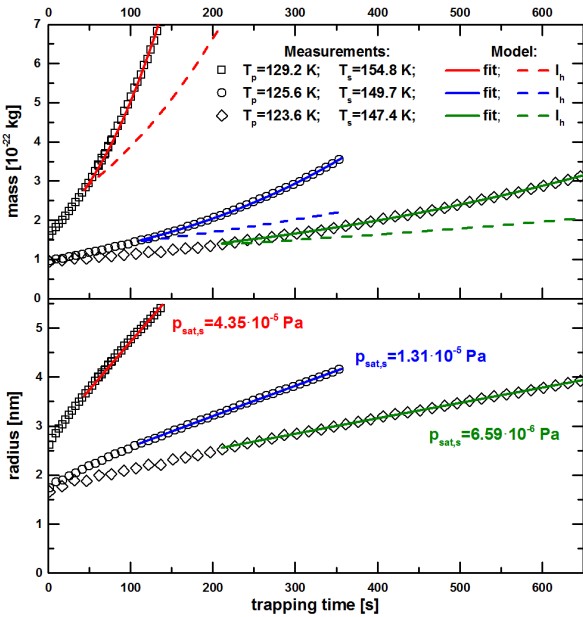

**Figure A1: Particle mass (upper panel) and radius (lower panel) as a function of trapping time in MICE for three exemplary measurements with sample surface temperatures $T_s$ of 147.4 K (◊), 149.7 K (○) and 154.8 K (□). The particle temperatures $T_p$ are between 123 K and 130 K assuring saturations above 1000. The green, blue and red curves show the results of numerically fitting**

**$p_{sat,s}$ in Eq. (A2) to the data. The dashed coloured lines show expected growth curves when assuming hexagonal ice $I_h$.**

The densities of ASW and hexagonal ice are very similar at temperatures under investigation (Brown et al., 1996; Loerting et al., 2011) so that the nature of the deposited phase does not enter in the calculation. We assume spherical particles with the above densities for the nucleus and the density of hexagonal ice for the water adsorbate to calculate the particle radius as function of time. The results are shown in the lower panel of Fig. A1. The green, blue and red curves represent numerical fits

of $p_{sat,s}$ in Eq. (A2) (numbers in the lower panel). For comparison, the results of model runs assuming the vapor pressure of hexagonal ice (Murphy and Koop, 2005) are shown by the dashed lines. The vapor pressure of the investigated ice phase is according to these curves significantly higher than the one of hexagonal ice. Fit uncertainties of $p_{sat,s}$ were typically on the order of 1%. The data were evaluated using Eq. (A2) only after the particles have gained at least 3 monolayers of $H_2O$ to avoid a possible influence of the particle material on the sublimation rate. The results reported in this work are indeed

independent of the type and initial radius (1.6 nm – 2.5 nm) of the particles under investigation. Using one fit parameter only ($p_{sat,s}$), the implemented growth model represents the data at all temperatures very well.

In this work, only spherical nuclei and ice particles are considered. However, NLCs form under conditions investigated in this work and light scattering models showed better agreement to NLC data retrieved by satellite and LIDAR remote sensing instruments when analysed under the assumption of aspherical ice particle shapes (Eremenko et al., 2005; Hervig et al.,

2012; Kiliani et al., 2015). At the particle temperatures investigated here (below $T_p$=130 K) water is most likely deposited as ASW onto the ice particles, which makes aspherical particle growth unlikely. In addition, the growth model fit does not require a changing aspect ratio to achieve very good agreement with the measured data excluding an increasing aspect ratio with particle growth. On the basis that metal oxide nanoparticle produced in similar arrangements have been shown to be compact and spherical (Giesen et al., 2005; Janzen et al., 2002; Nadeem et al., 2012), we estimate the maximum relative

uncertainty due to non-sphericity of the ice particles to 5 %. The main uncertainty in $p_{sat,s}$ is caused by the uncertainty in $T_s$ which is between 0.2 K and 0.4 K depending on the applied conditions.



At $T_w$ =135 K the time needed for a 2 nm radius particle to grow to a radius of about 5 nm is several hours. At 160 K the time is of the order of seconds only. These very slow and very fast growth rates set the experimental temperature limit for sublimation rate measurements with this setup.

**Appendix B: Relative non-isothermal vapor pressure measurements using an ionization gauge (T=166 K – 190 K)**

In order to extend the saturation vapor pressure measurements to temperatures above 160 K, an additional setup to measure the relative vapor pressure difference between ice deposited below 160 K and hexagonal ice was built. A schematic representation of the experimental setup is depicted in Fig. B1.

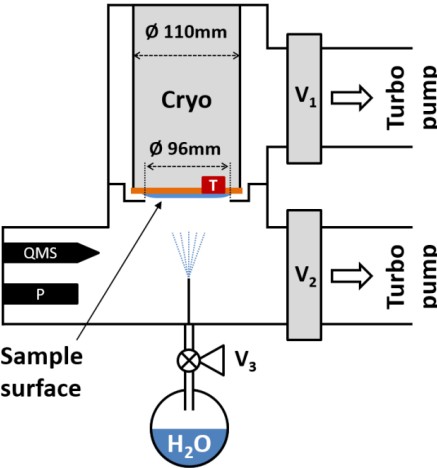

**Figure B1: Experimental setup used for the relative saturation vapor pressure measurements between 166 K and 190 K. A hot-
cathode ionization gauge (P) is employed in a vacuum chamber which is evacuated via valves $V_1$ and $V_2$ by two turbo molecular pumps. Simultaneous quadrupole mass spectrometer (QMS) measurements ensure that no other trace gases than $H_2O$ bias the pressure readout. See text for details.**

The setup consists of two interconnected vacuum chambers with a base residual gas pressure below $5 * 10^{-9}$ mbar. Upper and lower chamber can be evacuated via independent valves $V_1$ and $V_2$ by two turbo molecular pumps (Leybold Turbovac
350i, 290ls$^{-1}$ and Agilent Varian V 300HT, 250ls$^{-1}$). A closed cycle Helium cryostat (Advanced Research Systems, DE110 with GMX-20B) is mounted in the upper chamber with the cold sample surface pointing towards the lower chamber. The sample surface is a flat and polished copper disc with a diameter of 110 mm and with a Pt-100 temperature sensor attached to the side of the disc. A kinked separator ring is mounted between both chambers with an inner opening of Ø=96 mm and 2 mm distance between ring and cryostat. The ring serves as a barrier for water molecules from the lower to the upper chamber
during the experiment. Water vapor is provided from a flask containing Nanopure™ water that has been subject to several freeze-pump-thaw cycles to remove dissolved gases from the liquid prior to deposition. The water reservoir is connected to the vacuum chamber via the fine dosing valve $V_3$ and a thin tube such that after opening the valve a deposition rate of about 8 nm s$^{-1}$ on the probe is obtained.

We pursued two methods for depositing water vapor onto the sample surface: a) Nano-crystalline ice is produced using the
same procedure as with the MICE-TRAPS setup, either via deposition of ASW at 100 K followed by crystallization during warm-up or by direct deposition at 150 K resulting in a roughly 15 µm thick ice film. Both chambers were being evacuated during deposition. b) To create hexagonal ice, the fine dosing valve was opened to full extend with $V_1$ and $V_2$ closed and while cooling the sample surface with 2 K min$^{-1}$ starting from 277 K. At about 269 K condensation of liquid water droplets could be observed by eye through a glass window mounted on the lower chamber. Sudden freezing of the water droplets was
observed at about 260 K and we expect the formed ice to be hexagonal ice at this temperature. Valve $V_3$ to the water reservoir is closed immediately after crystallization and the probe temperature is further decreased with about 3 K min$^{-1}$



down to 150 K with $V_1$ and $V_2$ being opened at about 210 K. The cooling is turned off at 150 K to allow for a slow sample warm-up (<0.5 K min$^{-1}$). From this point on, the measurement procedure was identical for both deposition methods.

The temperature of the sample disc was measured with a Pt-100 temperature sensor and a distributed set of 6 Si-diode sensors. It was found, that during warm-up the sample surface temperature is homogeneous to within 0.2 K and the absolute

uncertainty of the temperature measurement was estimated to be 0.5 K. During warm-up, $V_2$ is closed to reduce water vapor loss by pumping. The vapor pressure of the deposited ice phase was measured as function of the sample temperature with a hot cathode ionization gauge (P; Oerlikon Leybold Ionivac ITR 90). The ITR 90 is a combined instrument comprised of a Pirani sensor for higher pressures and a Bayard Alpert hot cathode ionization sensor for lower pressures. Below 5.5 ∗ 3 mbar, which is the case for all measurements presented here, only the hot cathode ionization sensor is active. The sensor has

a characteristic curve calibrated for $N_2$ and the pressure measured by the device can be obtained via RS232 interface. All data in this work obtained using the ITR 90 is presented as recorded from the device without additional data processing.

Simultaneous residual gas measurements with a Quadrupole Mass Spectrometer (QMS; Peiffer Prisma Plus QMA-200) ensured that no significant amount of trace gases other than $H_2O$ bias the recorded total pressure readout. The resulting unprocessed recorded data are shown in Fig. B2.

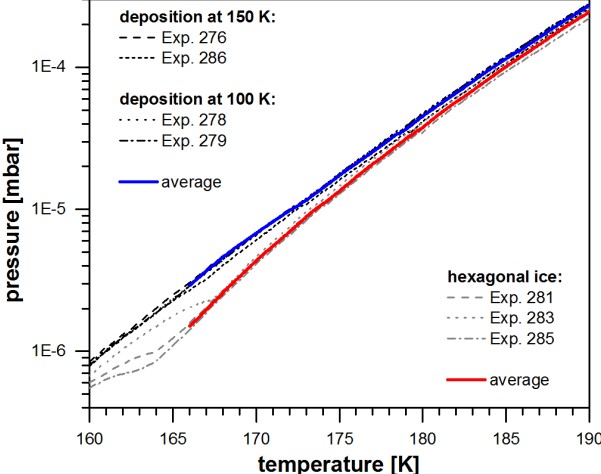

**Figure B2: Vapor pressure between 160 K and 190 K after deposition at 100 K and 150 K (black lines, 4 runs) and after crystallization of hexagonal ice from liquid water at 260 K (grey lines, 3 runs). The solid blue and red lines represent calculated mean values for deposition at 100 K / 150 K and hexagonal ice, respectively.**

In total seven measurements have been performed: 4 times after deposition at 100 K or 150 K (black lines) and 3 times after

deposition of hexagonal ice (grey lines) as described above. All four measurements of water ice deposited at 150 K and 100 K are very close, i.e. crystalline ice deposited at 150 K exhibits the same vapor pressure as ice crystallized after deposition of ASW at 100 K. This indicates, that at these temperatures, independent of deposition temperature, all four samples consist of the same ice polymorph.

For hexagonal ice all curves fall onto each other above 168 K showing a decreasing difference towards the ices deposited at

and below 150 K. Below 168 K, the three measurements of hexagonal ice show deviations, which can be explained by the following: During cool-down residual water desorbing from the inner surfaces of the vacuum chamber deposits onto the hexagonal ice film forming a layer of the same ice that is created when depositing water directly at 150 K. After some time of pumping and sample temperature increase, the residual water source is depleted and the layer on top of the hexagonal ice film begins to evaporate. Eventually, the over-layer will evaporate completely and expose the hexagonal ice below. The

transition from one ice phase being exposed at the surface to the other can be seen in all three measurements of hexagonal ice in Fig. B2. Therefore, the analysis of the data is restricted to temperatures above 166 K. Depending on ice thickness, all





ice is evaporated somewhat above 190 K, which limits our data to temperatures between 166 K and 190 K. Absolute vapor pressure measurements with the accuracy required to distinguish between different ice phases at such low temperatures are difficult to achieve with this setup. However, the measurements were reproducible and we can directly compare the vapor pressure of ices deposited below 160 K with hexagonal ice, relying for the latter on the accuracy of the well-established

parameterization by Murphy and Koop (Murphy and Koop, 2005). In this way, we avoid many uncertainties and systematic errors occurring in absolute vapor pressure measurements. We calculated the mean and standard deviation for all runs of low temperature vapor deposited ice between 166 K and 190 K (blue curve). For hexagonal ice, we use experiment 281 and 285 between 166 K and 169 K and all three runs above 169 K (red curve). The recorded vapor pressures were highly reproducible and the ratio of the vapor pressures of the two ice phases could be determined with an accuracy of 10 %.

**Author contribution**

MN and DD designed the experiments. MN carried out the MICE-TRAPS experiments. MN and DD carried out the pressure gauge experiments. MN performed the data analysis. MN prepared the manuscript with contributions from all co-authors. DD and TL supervised the experiments.

**Competing interests**

The authors declare that they have no conflict of interest.

**Acknowledgements**

The authors thank the German Federal Ministry of Education and Research (BMBF, grant number 05K13VH3 and 05K16VHB) and the German Research Foundation (DFG, grant number LE 834/4-1) for financial support of this work. We acknowledge support by Deutsche Forschungsgemeinschaft and Open Access Publishing Fund of Karlsruhe Institute of
Technology.

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
