# Peer review of "The vapor pressure over nano-crystalline ice"

_Atmospheric Chemistry and Physics, 2017_

## Referee Comment (RC1) · Anonymous Referee #1 · 14 Dec 2017

This manuscript (acp-2017-1101) reports temperature dependent measurements of the vapor pressure over nano-crystalline ice. The measurements were performed in two manners. In the first manner, the growth of seed nanoparticles in the MICE-TRAPS (molecular flow ice cell within the trapped reactive atmospheric particle spectrometer) exposed to a supersaturation of water vapor was used to infer the saturation vapor pressure over the ice sample surface in the cell. This approach allowed investigation of vapor pressures from 135 K to 160 K. The second approach to measure vapor pressure utilized a relative non-isothermal vapor pressure approach where ionization gauge pressure as a function of temperature gave the vapor pressure. This approach was applied from 166 K to 190 K. Combined, the authors found that at low temperatures (<160 K), where nano-crystalline ice is metastable for long time periods, saturation vapor pressures were 100% to 200% higher than for that of hexagonal ice, whereas at higher temperatures (>160 K) saturation vapor pressures were at most 18% higher

than that of hexagonal ice. The higher saturation vapor pressure at lower temperatures is described quantitatively by the Kelvin equation, assuming a nano-grain diameter of 7-19 nm.

This manuscript is clearly and concisely written with very clear figures. The experimental approach is novel and carefully performed. The results reported are significant, are very nicely put into the context of previous work on the subject, and will be very useful to the community. The subject of the manuscript is within the scope of Atmospheric Chemistry & Physics and should be published subject to addressing the minor comment below.

Comment:

In the revised manuscript, the authors should ensure that all abbreviations are defined. For example, in Appendix A, "k" in Eq. A2 is presumably the Boltzmann constant, though it is not defined. Similarly, "Tw" is not defined. On page 10, line 17, "NLC" presumably refers to noctilucent clouds but is not defined.

---

## Referee Comment (RC2) · Anonymous Referee #2 · 15 Dec 2017

The vapor pressure over nano-crystalline ice

This is a generally solid manuscript that describes new measurements of the vapor pressure of ice at low temperatures. I recommend publication with some revisions.

The most important is that in a number of places in the manuscript that language should be changed to say that the data are consistent with nano-crystalline ice, not that the data support or prove nano-crystalline ice. There are no data to directly show that the ice contained nano crystals. There are other possible explanations for the vapor pressure being elevated over that of hexagonal or stacking-disordered ice. In particular, some disorder within the stacking planes, as well as between them, could easily account for the change in Gibbs energy. Or the vapor pressure difference could indeed be from nano-crystals. Without direct evidence, the language in the paper is too certain of one explanation. This is especially true for the abstract but occurs elsewhere

as well. I think that Figure 3 could be eliminated and replaced with a short calculation: the crystal sizes inferred from the vapor pressure difference are consistent with the previous literature.

Something that could be made clear is that the ionization gauge is measuring a pressure that is different than the vapor pressure. In free molecular flow the water partial pressure in the warm part of the chamber near the gauge is not the same as the partial pressure above the sample, but rather differs by a factor of sqrt(T). It is only by normalizing to the vapor pressure of hexagonal ice with the same temperature gradients in the chamber that the correct measurement is made. This is rather vague in the manuscript/supplemental material.

Two questions where I am curious, not necessarily requiring changes:

I'm curious about the stability of nano-crystalline ice. Surely there would be a distribution of crystal sizes. If the vapor pressure is controlled by surface curvature, then there would be a Bergeron process and the larger crystals would grow at the expense of smaller ones, and the vapor pressure would slowly decrease.

I'm curious why, with both a residual gas analyzer and an ionization gauge available, the authors chose to use the ionization gauge to monitor the water vapor rather than the RGA water signal.

---

## Author Comment (AC1) · 1 Feb 2018

**Response to comments of the Anonymous Referee 1 on the manuscript entitled "The vapor pressure over nano-crystalline ice"**

We thank the reviewer for the very encouraging review and the thoughtful comments which we address individually below:

**Comment 1:** *In the revised manuscript, the authors should ensure that all abbreviations are defined. For example, in Appendix A, "k" in Eq. A2 is presumably the Boltzmann constant, though it is not defined. Similarly, "Tw" is not defined. On page 10, line 17, "NLC" presumably refers to noctilucent clouds but is not defined.*

**Response:** We thank the Referee for the thorough reading of our manuscript uncovering undefined abbreviations. We carefully revised the manuscript to make sure that no abbreviations are undefined any more.

**Changes made:**

- Page 9, line 31: … particles, ***k is the Boltzmann constant***, $v_{th,s} = \sqrt{8k\boldsymbol{T_s}/\pi m_{H_2O}}$ is…
- Page 10, line 17: However, ***noctilucent clouds*** form under…
- Page 11, line 1: At $\boldsymbol{T_s}$ =135 K the time needed for…

---

## Author Comment (AC2) · 1 Feb 2018

**Response to comments of the Anonymous Referee 2 on the manuscript entitled "The vapor pressure over nano-crystalline ice"**

We thank the reviewer for the very encouraging review and the thoughtful comments which we address individually below:

**Comment 1:** *The most important is that in a number of places in the manuscript that language should be changed to say that the data are consistent with nano-crystalline ice, not that the data support or prove nano-crystalline ice. There are no data to directly show that the ice contained nano crystals. There are other possible explanations for the vapor pressure being elevated over that of hexagonal or stacking-disordered ice. In particular, some disorder within the stacking planes, as well as between them, could easily account for the change in Gibbs energy. Or the vapor pressure difference could indeed be from nano-crystals. Without direct evidence, the language in the paper is too certain of one explanation. This is especially true for the abstract but occurs elsewhere as well.*

**Response:** We agree with the Referee that our measurements do not represent direct evidence for the nano-crystalline nature of the ice polymorph crystallized from ASW below 160 K. We made modifications to several text passages in the manuscript emphasizing that our data is consistent with the well-supported assumption of nano-crystals rather than a proof for nano-crystalline ice. The mean crystal size calculated using our data is in very good agreement with literature results. In addition to the above mentioned changes, we added a short paragraph on the potential influence of stacking disorder and defects on the vapor pressure of the crystalline ice polymorph.

**Changes made:**
- Page 1, line 12-13: Here, we present laboratory measurements on the saturation vapor pressure over **ice crystallized from ASW (deleted "nano-crystalline ice")** between 135 K and 190 K. Below 160 K, where **crystallization of ASW is known to form** nano-crystalline ice, we obtain…
- Page 1, line 19-23: Our measurements **are consistent with the assumption**, that (…) nano-crystalline ice **with mean diameter between 7 nm and 19 nm** forms thereafter by crystallization within the ASW matrix. **The estimated crystal sizes are in agreement with reported crystal size measurements** and remain stable for hours below 160 K. Thus, **this ice polymorph (deleted "nano-crystalline ice")** may be regarded as an independent phase for many atmospheric processes below 160 K and we parameterize its vapor pressure **(deleted "of nano-crystalline ice")** using a constant Gibbs free energy difference (…)
- Page 2, line 5: we deleted "**nano-crystalline**"
- Page 7, line 22-26: **Stacking disorder in ice $I_{sd}$ is expected to contribute to the free energy difference $\Delta G_{sd \to h}$ with less than 10 J mol$^{-1}$ (Hondoh et al., 1983; Hudait et al., 2016). The energy contribution of stacking faults therefore is not high enough to explain the variations in measured Gibbs free energy differences $\Delta G_{sd \to h}$ of 20 J mol$^{-1}$ to 180 J mol$^{-1}$ of ice $I_{sd}$ at temperatures above 180 K. Defects beyond stacking faults are proposed to explain the observed energy difference of up to 180 J mol$^{-1}$ (Hudait et al., 2016). However, it is unlikely that defects make up for an energy difference in the order of 1 kJ mol$^{-1}$ as observed in this study below 160 K. We therefore conclude that an increase of defects beyond stacking faults below 180 K is not the major process causing the observed elevated vapor pressure.** In order to calculate crystal diameters, we assumed that the crystallites are composed of ice $I_{sd}$ and that this

ice polymorph is described by a temperature independent Gibbs free energy difference $\Delta G_{sd \to h}$ of 20 J mol$^{-1}$ to 180 J mol$^{-1}$. An increase of defects beyond stacking faults in the ice I$_{sd}$ polymorph with decreasing temperature might **still** cause a **small** increase in $\Delta G_{sd \to h}$ , **which would lead to a change in calculated crystallite sizes.** (…)

- Page 8, line 8-9: Since deposition between 140 K and 160 K as well as crystallization of ASW deposited at 95 K and 100 K leads **to identical vapor pressures (deleted "the same nano-crystallite sizes"),** it is very likely that ice deposition up to 160 K proceeds by an initial deposition of ASW followed by rapid crystallization **(deleted "to nano-crystallite sizes")**.
- Page 8, line 22-23: (…) has no influence on the **crystallized ice polymorph (deleted " ice grain sizes formed during crystallization)**.
- Page 8, line 34-36: The observed high vapor pressure **can be** quantitatively explained with the high surface energy to volume energy ratio of nano-scale crystallites (Kelvin effect). A transition in the vapor pressure data above 165 K **is consistent** with the thermally activated relaxation of (…).

**Comment 2:** *I think that Figure 3 could be eliminated and replaced with a short calculation: the crystal sizes inferred from the vapor pressure difference are consistent with the previous literature.*

**Response:** Figure 3 illustrates the compelling agreement of measured grain diameters from independent studies with grain diameters inferred from our vapor pressure data. In all studies shown, crystal diameters where determined in ice crystallized from amorphous ices. We therefore consider Figure 3 to be very important for the line of argument of the manuscript, which is based on nano-crystallites being the most likely source of the observed enhanced vapor pressure. Thus, we would rather keep Figure 3 in the manuscript.

**Changes made: -**

**Comment 3:** *Something that could be made clear is that the ionization gauge is measuring a pressure that is different than the vapor pressure. In free molecular flow the water partial pressure in the warm part of the chamber near the gauge is not the same as the partial pressure above the sample, but rather differs by a factor of sqrt(T). It is only by normalizing to the vapor pressure of hexagonal ice with the same temperature gradients in the chamber that the correct measurement is made. This is rather vague in the manuscript/supplemental material.*

**Response:** We were well aware of that fact, it is one of the reasons to give relative vapor pressures only. Nevertheless we added this information to the experimental part in the Appendix.

**Changes made:**

- Page 12, line 11: **The data shown thus deviates from the vapor pressure above the sample surface by the H$_2$O calibration curve of the sensor. In addition, in free molecular flow the partial pressure measured in the warm part of the chamber near the gauge ($T_w$) differs from the partial pressure above the cold ice sample surface ($T_c$) by a factor $\sqrt{T_c/T_w}$.**
- Page 13, line 3-4: we can directly compare the **unprocessed recorded** vapor pressure…

**Two questions where I am curious, not necessarily requiring changes:**

**Comment 4:** *I'm curious about the stability of nano-crystalline ice. Surely there would be a distribution of crystal sizes. If the vapor pressure is controlled by surface curvature, then there would be a Bergeron process and the larger crystals would grow at the expense of smaller ones, and the vapor pressure would slowly decrease.*

**Response:** We also assume, that there is a distribution of crystal sizes and that the larger crystals grow according to the Bergeron process. Below 160 K, however, crystal growth is too slow to cause a significant change in vapor pressure on time scales (1day) of our experiment. Above 160 K, crystal growth speeds are high enough to be observed in our pressure gauge experiment. This is supported by our data as well as the study of Hansen et. al. (2008).

**Comment 5:** *I'm curious why, with both a residual gas analyzer and an ionization gauge available, the authors chose to use the ionization gauge to monitor the water vapor rather than the RGA water signal.*

**Response:** At temperatures of about 170 K, the $H_2O$ signal of our residual gas analyzer (RGA) began to saturate. Thus, $H_2O$ pressure measurements of the RGA were only valid in a very narrow temperature range between 166 K and somewhat below 170 K. Nevertheless, in this temperature range the comparison of RGA signals of crystallized ASW and hexagonal ice showed the elevated vapor pressure of the crystallized ASW as well. In order to keep the discussion of the results concise, we decided to show the results obtained with the ionization gauge only, which were valid up to 190 K.

**Changes made:**
- Page 12, line 13: ***However, the data recorded by the QMS was not used to evaluate the water vapor partial pressure in this work as the QMS signal on m/q channel 18 saturated at a temperature of about 170 K.***
- Page 12, line 13: The resulting unprocessed recorded data ***of the ionization gauge*** are shown (…)